# Fungal and Oomycete Diseases of Minor Tropical Fruit Crops

**Latiffah Zakaria**

School of Biological Sciences, Universiti Sains Malaysia, USM, Gelugor 11800, Penang, Malaysia; lfah@usm.my

**Abstract:** Minor tropical fruits are grown on a small scale and provide income to smallholder farmers. The cultivation of these fruit crops indirectly contributes to the economy of producing countries as well as to food and crop security. Dragon fruits, guava, passionfruit, lychee, longan, mangosteen, durian, and rambutan are common minor fruit crops. In recent years, the international trade of some of these minor tropical fruits, particularly dragon fruit, passionfruit, guava, and lychee, has increased due to their nutritional value, with various health benefits. Similar to other crops, minor fruit crops are susceptible to fungal and oomycete diseases. These diseases negatively affect the yield and quality of fruit crops, leading to substantial losses. In this context, the knowledge of disease types and causal pathogens is fundamental to develop suitable disease management practices in the field as well as appropriate post-harvest treatments.

**Keywords:** fungi and oomycete; diseases; dragon fruits; guava; passionfruit; lychee; longan; mangosteen; durian; rambutan

## 1. Introduction

Minor tropical fruits are cultivated on a small scale and traded in small capacity, mostly at the local or regional levels. The consumption of these fruits is often limited to the local population [1]. Many minor tropical fruits, such as dragon fruits, guava, passionfruit, lychee, longan, mangosteen, durian, and rambutan, contribute to the economy of their respective producing countries. Nowadays, the production and trade of these minor tropical fruits have been rapidly expanding owing to their health benefits, and these crops have thus garnered much importance in the global market [2].

In many producing countries, minor tropical fruits have become a source of income, particularly for rural smallholder farmers. According to a survey by the FAO, earning from the cultivation and production of minor tropical fruits can contribute up to 75% of the total income. This may be due to the higher wholesale prices of these fruit crops, reaching around USD 4 per kilogram for lychee and guava and USD 13 per kilogram for durian, mangosteen, and passion fruit. Thus, minor tropical fruit cultivation is very profitable, provided that other factors, such as the handling and transportation of fruits, are well managed and cost effective [2].

In international markets, particularly in European countries and the US, minor tropical fruits are used primarily by Asian communities or ethnic markets and often by premium retailers for dietary and health preferences [2]. Familiar or popular minor tropical fruits in the international market include guava, passion fruit, and lychee. Additionally, some fruits, including dragon fruit, mangosteen, and rambutan, have been recognized as superfoods.

During 2015–2017, Asian countries were responsible for approximately 86% of minor tropical fruit production. The largest producers include India (24%) and China (22%), where these fruits are mainly available in local markets. Other major producing countries include Southeast Asian countries, including Thailand, Vietnam, Indonesia, and Malaysia, as well as South American countries, particularly Brazil [2].

Similar to other tropical fruit crops, however, diseases represent the most important constraint to minor tropical fruit crop production. Nearly all minor tropical fruit crops are infected by one or more plant pathogenic fungi and oomycetes, which are fungal-like organisms (Table 1). Thus, the knowledge of various diseases of these fruit crops is important to determine optimal cultivation regions and practices, post-harvest treatments, trade (market type or retail), and sustainability and profitability of their production [3]. In this context, the present review highlights the common fungal and oomycete diseases of popular minor tropical fruit crops, including dragon fruit, passion fruit, longan, lychee, rambutan, mangosteen, and durian.

**Table 1.** Fungal and oomycete diseases associated with minor tropical fruit crops reported in several countries.

| | Dragon Fruit (*Hylocereus* spp.) | | |
|---|---|---|---|
| **Disease** | **Causal Pathogen/** *Hylocereus* **spp.** | **Country** | **References** |
| Anthracnose (fruit and stem) | *Colletotrichum gloeosporioides sensu lato* (stem of *H. undatus*; stem of *H. megalanthus*; stem and fruit of *Hylocereus* spp.) | Miami-Dade County, Florida, USA; Brazil, Malaysia | Palmateer et al. [4], Takahashi et al. [5], Masyahit et al. [6] |
| | *Colletotrichum gloeosporioides* (young stem and fruit of *H. undatus*) | China, Taiwan | Ma et al. [7], Lin et al. [8] |
| | *Colletotrichum siamense* (stem and fruit of *H. undatus*; stem of *H. polyrhizus*) | Thailand, China | Meetum et al. [9], Zhao et al. [10] |
| | *Colletotrichum karstii* (stem of *H. undatus*) | Brazil | Nascimento et al. [11] |
| | *Colletotrichum fructicola* (stem of *H. undatus* and *H. monacanthus*) | the Philippine | Evallo et al. [12] |
| | *Colletotrichum truncatum* (fruit of *H. undatus*, stem of *H. polyrhizus*) | Malaysia, China | Guo et al. [13], Iskandar Vijaya et al. [14] |
| | *Colletotrichum aenigma* (stem and fruit of *H. undatus*) | Thailand | Meetum et al. [9] |
| Stem lesion/spot | *Colletotrichum siamense* (stem of *H. undatus*) | Andaman Islands, India | Abirami et al. [15] |
| | *Curvularia lunata* (stem of *H. polyrhizus*) | Malaysia | Masratul Hawa et al. [16] |
| Storage fruit rot | *Gilbertella persicaria* (*H. costaricensis*) | China | Guo et al. [17] |
| Fruit blotch and stem rot | *Bipolaris cactivora* (*H. undatus*) | South Florida, Israel, Thailand, Vietnam | Tarnowski et al. [18], Ben-Ze'ev et al. [19], He et al. [20], Oeurn et al. [21] |
| Stem blight | *Alternaria* sp. (*H. undatus*) | South Florida, USA | Patel and Zhang [22] |
| Post-harvest disease | *Alternaria alternata* (*H. undatus*) | Brazil | Castro et al. [23] |
| Stem and Fruit Spot | *Aureobasidium pullulans* (*Hylocereus* spp.) | China | Wu et al. [24] |

| Disease | Causal pathogen | Country | References |
|---|---|---|---|
| Stem blight | *Sclerotium rolfsii* (*H. undatus*) | China | Zheng et al. [25] |
| Stem reddish brown spot | *Nigrospora sphaerica* (*H. undatus*) | China | Liu et al. [26] |
| Stem reddish brown spot | *Nigrospora lacticolonia* and *N. sphaerica* (*H. polyrhizus*) | Malaysia | Kee et al. [27] |
| Stem canker, black rot, brown spot, fruit internal browning, fruit canker | *Neoscytalidium dimidiatum* (*H. undatus* and *H. monacanthus*) | Israel, Taiwan, Malaysia, China; Florida, USA; Puerto Rico | Chuang et al. [28], Lan et al. [29], Ezra et al. [30], Masratul Hawa et al. [31], Yi et al. [32], Sanahuja et al. [33], Serrato-Diaz and Goenaga [34] |
| Stem gray blight | *Diaporthe arecae, Diaporthe eugeniae, Diaporthe hongkongensis, Diaporthe phaseolorum,* and *Diaporthe tectonendophytica* (*H. polyrhizus*) | Malaysia | Huda-Shakirah et al. [35] |
| | *Diaporthe ueckerae* (*H. polyrhizus* and *H. undatus)* | Taiwan | Wang et al. [36] |
| | *Fusarium proliferatum, Fusarium fujikuroi* (*H. polyrhizus*) | Malaysia | Masratul Hawa et al. [37], [38] |
| Stem rot | *Fusarium solani* (*Hylocereus* sp.) | Bali, Indonesia Banyuwangi Regency, Indonesia | Rita et al. [39], Sholihah et al. [40] |
| | *Fusarium* sp. (*Hylocereus* sp.) | Lombok Utara and Central Bangka Regency, Indonesia | Isnaini et al. [41], Kurniasari et al. [42] |
| | *Neocosmospora rubicola/F. solani* species complex (*H. costaricensis*) | Dongfang, Hainan Province, China | Zheng et al. [43] |
| Basal rot | *Fusarium oxysporum* (*H. undatus*, *Selenicereus megalanthus*, *H. polyrhizus*) | Gran Buenos Aires, Argentina; Colombia, Bangladesh | Wright et al. [44], Salazar-González et al. [45], Mahmud et al. [46] |
| Stem blight | *Fusarium oxysporum* (*H. polyrhizus*) | Malaysia | Mohd Hafifi et al. [47] |
| | *Fusarium lateritium, Fusarium semitectum* | Vietnam | Le et al. [48] |
| Fruit rot | *Fusarium oxysporum, Fusarium dimerum* (*H. undatus*) | Shanghai, China | Zhi-Jing et al. [49] |
| | *Fusarium dimerum, Fusarium equiseti* (*H. undatus*) | Mekong, Delta, Vietnam | Ngoc et al. [50] |

| Guava (*Psidium guajava* **L.**) | | | |
|---|---|---|---|
| **Disease** | **Causal pathogen** | **Country** | **References** |
| Fusarium wilt | *Fusarium oxysporum* f. sp. *psidii* | India | Prasad et al. [51], Misra and Gupta [52] |
| | *Fusarium oxysporum, Fusarium solani* | India | Misra and Pandey [53] |

| Disease | Causal Pathogen | Country | References |
|---|---|---|---|
| Decline | *Fusarium proliferatum, Fusarium chlamydosporum* | India | Misra and Gupta [52], Gupta and Misra [54] |
| | *Fusarium oxysporum* f.sp. *psidii, Fusarium solani* | India | Dwivedi and Dwivedi [55], Misra et al. [56,57], Misra [58] |
| | *Fusarium oxysporum* f. sp. *psidii, Fusarium falciforme* | India | Gangaraj et al. [59] |
| | *Fusarium oxysporum* f.sp. *psidii, Fusarium solani* f.sp. *psidii* (no information on the nematode) | District of Punjab, Pakistan | Aftab [60] |
| | *Fusarium solani* (*Meladogyne mayaguensis*) | Brazil | Gomes et al. [61] |
| | *Fusarium oxysporum* (*Meloidogyne incognita*) | Haryana, India | Madhu et al. [62] |
| | *Fusarium oxysporum* f.sp. *psidii* (*Meloidogyne enterolobii*) | Ratlam district, India | Singh [63] |
| | *Fusarium solani* (*Meloidogyne enterolobii*) | Brazil | Veloso et al. [64] |
| | *Fusarium solani, Fusarium oxysporum* (no information on the nematode) | Pakistan | Khizar et al. [65] |
| Anthracnose | *Colletotrichum gloeosporioides* complex | Italy | Weir et al. [66] |
| | *Colletotrichum siamense* complex | India and Mexico | Sharma et al. [67], Rodríguez-Palafox et al. [68] |
| | *Colletotrichum abscissum, Colletotrichum simmondsii* | Brazil | Bragança et al. [69], Cruz et al. [70] |
| | *Colletotrichum guajavae* | India | Damm et al. [71] |
| Crown rot | *Fusarium verticillioides* | India | Sanjeev and Brijpal [72], Baloch et al. [73] |
| | *Aspergillus fumigatus, Aspergillus niger, Aspergillus tamarii, Aspergillus japonicus, Aspergillus flavus* | Phillipine | Valentino et al. [74] |
| Fruit rot | *Fusarium oxysporum* | Egypt, Nigeria | Mathew [75], Amadi et al. [76], Embaby and Korkar, [77], Mairami et al. [78] |
| | *Aspergillus awamori* | Pakistan | Akhtar et al. [79] |
| | *Phytophthora nicotianae* | Bangladesh | Pervez et al. [80] |
| | *Neoscytalidium dimidiatum* | Malaysia | Ismail et al. [81] |
| | *Lasiodioplodia theobromae* | Malaysia | Zee et al. [82] |
| Canker | *Diplodia natalensis, Pestalotia psidii* | India | Misra 2012 [83] |

| Passion Fruit (*Passiflora edulis* Sim.) | | | |
|---|---|---|---|
| **Disease** | **Causal Pathogen** | **Country** | **References** |
| Wilt | *Fusarium oxysporum* f. sp. *passiflorae* | Brazil, North America, Portugal, New Zealand, | Rooney-Latham et al. [84], Garcia et al. [85], Melo et al. [86], Thangavel et al. [87] |
| | *Fusrium oxysporum* | Iksan and Jeju, Korea | Joa et al. [88] |
| | *Fusrium solani* | Zimbabwe | Cole et al. [89] |

| Collar rot | *Fusarium incarnatum, Fusarium solani, Fusarium proliferatum* | Colombia | Henao-Henao et al. [90] |
|---|---|---|---|
| | *Fusarium nirenbergiae* | Italy | Aiello et al. [91] |
| | *Fusarium solani* f.sp. *passiflorae* (*Fusarium solani*) | Brazil, USA, China, Uganda | Emechebe et al. [92], Ploetz [93], Li et al. [94], Ssekyewa et al. [95], Bueno et al. [96], Marostega et al. [97], Zhou et al. [98] |
| Canker | *Fusarium solani* *Fusarium oxysporum* f.sp. *pasiflorae* | Florida, USA | Manicom et al. [99], Ploetz [100], Anderson and Chambers [101] |
| | *Fusarium solani* | Taiwan and Uganda (reported as Nectria canker) | Emecahebe et al. [92], Lin and Chang [102] |
| | *Fusarium solani* | Kenya | Wangungu et al. [103], Power and Verhoeff [104] |
| Dieback | *Fusarium oxysporum. subglutinans, Fusarium pseudoanthophilum, Fusarium solani, Fusarium. semitectum* | Kenya | Amata et al. [105] |
| Stem bulging | *Gibberella fujukuroi, Fusarium* sp. | Sri Lanka | Wanniarachchi et al. [106], Rajapaksha et al. [107] |
| Anthracnose | *Colletotrichum boninense, Colletotrichum boninense, Colletotrichum truncatum, Colletotrichum gloeosporioides, Glomerella* sp. | Florida | Tarnowski and Ploetz [108] |
| | *Colletotrichum boninense* | Brazil | Tozze Jr. et al. [109] |
| | *Colletotrichum queenslandicum* | Northern Territory, Australia | James et al. [110] |
| | *Colletotrichum brevisporum* | Fujian Province, China | Du et al. [111] |
| | *Colletotrichum truncatum* | China and Taiwan | Zhuang et al. [112] Chen and Huang [113] |
| | *Colletotrichum brasiliense* | China | Shi et al. [114] |
| | *Colletotrichum constrictum* | Yunnan, China | Wang et al. [115] |

| Lychee (*Litchi chinense* Sonn.) | | | |
|---|---|---|---|
| **Disease** | **Causal Pathogen/Plant Parts** | **Country** | **References** |
| Anthracnose | *Colletotrichum gloeosporioides sensu lato* | | Fitzell and Coates [116], Coates et al. [117] |
| | *Colletotrichum gloeosporioides* (immature fruit and asymptomatic flowers) | Mexico | Martinez-Bolanos et al. [118] |
| | *Colletotrichum fioriniae* (fruit) | China | Ling et al. [119] |
| | *Colletotrichum karstii* (leaves) | Guangxi, China | Zhao et al. [120] |
| Pepper spot | *Colletotrichum gloeosporioides sensu lato* (fruit) | Australia | Cooke and Coates [121], Anderson et al. [122] |
| | *Colletotrichum siamense* (fruit) | Taiwan, China | Ni et al. [123] Ling et al. [124] |

| | | | |
|---|---|---|---|
| Blight of leaf, panicle and fruit | *Alternaria alternata* | Bihar, India | Kumar et al. [125] |
| Fruit rot (Brown rot) | *Fusarium incarnatum* | Hainan, China | Guo et al. [126] |
| Downy blight | *Phytophthora litchi* | Taiwan, Southern China | Kao and Leu [127], Wang et al. [128] |

| Longan (*Dimocarpus longan* Lour.) | | | |
|---|---|---|---|
| **Disease** | **Causal Pathogen** | **Country** | **References** |
| Downy blight (young leaves, panicles, flowers and fruits) | *Phytophthora litchi* | Taiwan | Ann et al. [129] |
| Inflorescence wilt, vascular and flower necrosis | *Fusarium decemcellulare* | Puerto Rico | Serrato-Diaz et al. [130] |
| Fruit rot (Brown rot) | *Phytophthora palmivora* | Thailand | Kooariyakul and Bhavakul [131] |
| | *Lasiodiplodia theobromae* | Puerto Rico | Serrato-Diaz et al. [132] |
| | *Lasiodiplodia pseudotherobromae* | Thailand | Pipattanapuckdee et al. [133] |
| Pericarp browning | *Phomopsis longanae, L. theobromae* | China | Chen et al. [134] Sun et al. [135] |
| Dieback | *Lasiodiplodia hormozganensis, Lasiodiplodia iraniensis, Lasiodiplodia pseudotheobromae*, and *Lasiodiplodia theobromae* | Puerto Rico | Serrato-Diaz et al. [136] |
| Inflorescence blight | *Lasiodiplodia theobromae* | Puerto Rico | Serrato-Diaz et al. [132] |

| Durian (*Durio zibethinus* L.) | | | |
|---|---|---|---|
| **Disease** | **Causal Pathogen** | **Country** | **References** |
| Patch canker or stem canker, fruit rot, seedling dieback, foliar blight and root rot | *Phytophthora palmivora* | Malaysia, Indonesia, Thailand, Brunei, Vietnam | Pongpisutta and Sangchote [137], Lim [138], Lim [139], Sivapalan et al. [140], Tho et al. [141] |
| Stem rot | *Fusarium solani* and *L. pseudotheobromae* | Thailand | Chantarasiri and Boontanom [142] |
| Leaf blight | *Rhizoctonia solani* | Vietnam and Peninsular Malaysia | Thuan et al. [143], Lim et al. [144] |
| Foliar blight and Dieback | *Rhizoctonia solani* | Malaysia | Lim et al. [144] |
| Leaf spot | *Phomopsis durionis* | Thailand | Tongsri et al. [145] |
| Fruit rot | *Sclerotium rolfsii* | Malaysia | Lim and Kamaruzaman [146] |
| | *Colletotrichum gloeosporioides, Lasiodiplodia theobromae* | Thailand | Sangchote et al. [147] |
| | *Aspergillus* spp., *Penicillium* sp., *Fusarium equiseti* (secondary invaders or weak pathogens) | Brunei | Sivapalan et al. [148] |
| Durian decline | *Pythium vexans, Phytophthora palmivora* | Queensland, Australia and Indonesia | O'Gara et al. [149] |
| | *Phytophthora palmivora, Pythium cucurbitacearum, Pythium vexans* | Indonesia | Santoso et al. [150] |

| Root rot and canker lesion | *Phytophthora nicotianae* | Sabah, Malaysia | Bong [151] |
|---|---|---|---|
| Root and stem rot | *Pythium cucurbitacearum*, *Pythium vexans* (syn. *Phytopytium vexans*), *Pythium deliense* | Queensland, Australia; Malaysia, Thailand, Indonesia, Vietnam | Lim and Sangchote [152], Vawdrey et al. [153], Thao et al. [154] |

| Rambutan (*Nephelium lappaceum* L.) | | | |
|---|---|---|---|
| **Disease** | **Causal Pathogen** | **Country** | **References** |
| Fruit rot | *Botryodiplodia theobromae*, *Colletotrichum gloeosporioides*, *Gliocephalotrichum bulbilium*, *Pestalotiopsis* sp., *Phomopsis* sp., *Glomerella* sp. | Hawaii, Puerto Rico, Malaysia, Thailand and Sri Lanka, China | Farungsang et al. [155], Sivakumar et al. [156], Sangchote et al. [157], He et al. [158] |
| | *Lasmenia* sp., *Gliocephalotrichum* spp., *Pestalatiopsis virgatula* | Hawaii | Nishijima et al. [159], Keith [160] |
| | *Gliocephalotrichum bulbilium*, *Gliocephalotrichum simplex*, *Colletotrichum fructicola*, *Colletotrichum queenslandicum* | Puerto Rico | Serrato-Diaz et al. [161], Serrato-Diaz, et al. [162], Serrato-Diaz et al. [163] |
| | *Gliocephalotrichum bacillisporum* | Malaysia | Intan Sakinah and Latiffah [164] |
| Corky bark | *Dolabra nepheliae* | Malaysia, Hawaii, Puerto Rico and Honduras | Booth and Ting [165], Combs et al. [166], Rossman et al. [167] |
| Stem canker | *Dolabra nepheliae* | Hawaii, Puerto Rico and Honduras | Rossman et al. [168,169] |
| Corky bark and dieback | *Lasiodiplodia brasiliensis*, *L. hormozganensis*, *Lasiodiplodia iraniensis*, *Lasiodiplodia pseudotheobromae*, *Lasiodiplodia theobromae*, *Neofusicoccum batangarum*, *Neofusicoccum parvum* | Puerto Rico | Serrato-Diaz et al. [130] |
| Powdery mildew | *Oidium nephelii* | Sri Lanka, the Philippine, Thailand and Malaysia | Garcia [170], Coates et al. [171], Rajapakse et al. [172] |
| Inflorescence wilt, flower and vascular necrosis | *Fusarium decemcellulare* | Puerto Rico | Serrato-Diaz et al. [130] |
| Leaves necrosis of rambutan seedlings | *Pseudocercospora nephelii* | Brunei, Malaysia (Sabah and Selangor) | Peregrine et al. [173] |

| Mangosteen (*Garcinia mangostana* L.) | | | |
|---|---|---|---|
| **Disease** | **Causal Pathogen** | **Country** | **References** |
| Leaf blight | *Pestalotiopsis flagisettula* | Thailand, Malaysia, North Queensland, Australia, Hawaii | Lim and Sangchote [174], Keith and Matsumoto [175] |
| Brown leaf spots and blotches | *Pestalotiopsis* sp. | Hawaii | Keith and Matsumoto [175] |
| Diplodia fruit rot | *Diplodia theobromae* | Thailand | Lim and Sangchote [174] |
| Fruit rot | *Gliocephalotrichum bulbilium* | Guangzhou, China | Li et al. [176] |
| | *Gliocephalotrichum bulbilium*, *Graphium* sp. | Thailand | Sangchote and Pongpisutta [177] |

| | *Mucor irregularis* | Wujing Town, Shanghai | Wang et al. [178] |
|---|---|---|---|
| Black aril rot | *Lasiodiplodia theobromae* | Hawaii | Ketsa and Paull [179] |
| White aril rot | *Phomopsis* sp. | Hawaii | Ketsa and Paull [179] |
| Soft aril rot | *Pestalotiopsis* sp. | Hawaii | Ketsa and Paull [179] |
| Decline | *Lasidiplodia theobromae, Lasiodiplodia parva* | Bahia, Brazil | Paim et al. [180] |
| Brown root rot | *Phellinus noxius* | - | Lim and Sangchote [174] |
| Stem canker and die-back | *Pestalotiopsis* sp. | - | Lim and Sangchote [174] |
| Thread blight | *Marasmiellus scandens* | - | Lim and Sangchote [174] |

## 2. Fungal Diseases of Dragon Fruit

Dragon fruit (*Hylocereus* spp.) is a climbing cactus of the Cactaceae family. Due to the presence of scales or bracts on the surface of the fruit, it is often called pitaya or pitahaya meaning "scaly fruit" and has acquired the English moniker "dragon fruit." As dragon fruit flowers bloom only at night, they are also called the lady of the night, moonflower, belle of the night, and queen of the night [181].

Dragon fruit is presumably native to Mexico, Central America, and South America, specifically southern Mexico, the Pacific side of Guatemala and Costa Rica, and El Salvador [182]. Dragon fruit is now acclimatized to and cultivated in several countries in Central, South, and North America, including Mexico, Guatemala, Colombia, Costa Rica, Peru, Venezuela, and the United States (Florida, Hawaii, and southern California). In Asia, dragon fruit is widely cultivated in Vietnam, Thailand, Malaysia, Cambodia, Indonesia, the Philippines, India, and Taiwan [183], with Vietnam being the largest producer.

Only four species of *Hylocereus* are widely cultivated, depending on the country of origin: *Hylocereus undatus* (pink skin with white flesh), *Hylocereus monacanthus* (=*Hylocereus polyrhizus*; pink skin and red flesh), *Hylocereus costaricencis* (pink skin with violet-red flesh), and *Hylocereus megalanthus* (=*Selenicerus polyrhizus*; yellow skin with white flesh) [184].

### 2.1. Anthracnose

Various fungal pathogens affect dragon fruit crop. According to Balendres and Bengoa [185], 21 fungal species are associated with dragon fruit diseases. Among these, anthracnose is the most severe disease occurring on the fruit and stem. Anthracnose can occur in the field following harvest. Notable symptoms of anthracnose appear either on the fruit or the stem (Figure 1A,B), showing as reddish-brown irregular or round spots that later merge, enlarge, and turn into dark brown sunken lesions. In these sunken lesions, conidial masses appear, and the lesions are surrounded by chlorotic halos [11]. Although anthracnose has been reported in several producing countries, so far there is a lack of reports from Vietnam, Indonesia, and Sri Lanka even though these three countries are among the major producers of dragon fruits.

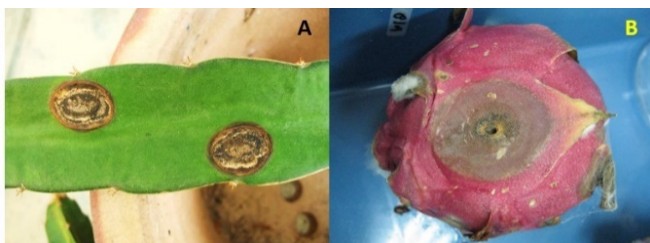

**Figure 1.** Anthracnose of dragon fruit. (**A**) Stem anthracnose. (**B**) Fruit anthracnose.

Before taxonomic revision of the *Colletotrichum* genus, *Colletotrichum gloeosporioides* was the most common species associated with dragon fruit anthracnose [4–6]. *Colletotrichum gloeosporioides* is a part of the *Colletotrichum gloeosporioides* species complex, comprising other species identified based on the molecular phylogeny of multiple genes [66]. Following taxonomic revision, *Colletotrichum gloeosporioides* was still identified as the causal pathogen of dragon fruit anthracnose in China and Taiwan [7,8]. Other species of *Colletotrichum* within the *Colletotrichum gloeosporioides* species complex causing dragon fruit anthracnose include *Colletotrichum siamense* [9,10], *Colletotrichum karstii* [11], and *Colletotrichum fructicola* [12]. In addition, *Colletotrichum truncatum* and *Colletotrichum boninense* have also been implicated in dragon fruit anthracnose [13,14] (Table 1).

### 2.2. Stem Rot

Stem rot of dragon fruit plants infected by *Fusarium* spp. has been reported in Malaysia and Indonesia (Table 1). Symptoms on infected *Hylocereus polyrhizus* (=*Hylocereus monacanthus*) include brown circular sunken lesions and etiological symptoms in the form of white mycelia and orange sporodochia. The causal pathogens include two *Fusarium* species, namely *Fusarium proliferatum* and *Fusarium fujikuroi* [37,38]. *Fusarium proliferatum* and *Fusarium fujikuroi* causing the stem rot of dragon fruit produce fumonisins, moniliformin, and beauvericin, and these mycotoxins contribute to the development and symptom expression of stem rot disease [186].

In Indonesia, the symptoms of stem rot include brown rot and wilting. *Fusarium solani* has been reported as the causal pathogen of stem rot in Bali [39] and Banyuwangi Regency [40]. Zheng et al. [43] reported *Hylocereus costaricensis* stem rot caused by *Neocosmospora rubicola* in Dongfang (Hainan Province, China).

Other stem rot diseases affecting dragon fruit include basal rot, stem blight, stem necrosis, stem canker, reddish brown spot, and stem gray blight (Table 1). Basal rot is caused by *Fusarium oxysporum* and has been reported in several dragon fruit species in Argentina, Colombia, and Bangladesh [44–46]. *Fusarium oxyporum* causes stem blight [47]. Stem necrosis is caused by *Curvularia lunata* [16] and stem canker by *Neoscytalidium dimidiatum* [37]. Two species of *Nigrospora*, namely *Nigrospora lacticolonia* and *Nigrospora sphaerica*, cause reddish brown spot [27]. Stem gray blight has been reported in Malaysia, and five *Diaporthe* species, namely *Diaporthe arecae*, *Diaporthe eugeniae*, *Diaporthe hongkongensis*, *Diaporthe phaseolorum*, and *Diaporthe tectonendophytica*, have been identified as the causal pathogens [35].

### 2.3. Fruit Rot

Anthracnose caused by *Colletotrichum* species is the most common post-harvest fruit rot of dragon fruit. In addition, *Bipolaris cactivora* and *Fusarium* spp. have been reported to cause fruit rot in dragon fruit. Infection often occurs in the presence of predisposing factors, such as wounds. According to literature, fruit rot caused *Bipolaris cactivora* is more prevalent than that caused by *Fusarium* spp. (Table 1). Pre- or post-harvest fruit rot has been reported in several countries, specifically in *Hylocereus undatus* [18–21,187,188]. Furthermore, *Fusarium oxysporum*, *Fusarium dimerum*, and *Fusarium equiseti* (Table 1) are associated with the fruit rot of dragon fruit [49,189]. Other fungal pathogens associated with dragon fruit disease are listed in Table 1.

## 3. Fungal Diseases of Guava

Guava (*Psidium guajava*) belongs to the myrtle family Myrtaceae and is widely distributed in tropical and subtropical regions of Asia, Africa, Oceania, and parts of the USA [190]. The origin of guava is not certain, but it is assumed that the plant originated from southern Mexico through Central and South America, probably from Mexico to Peru, as in these areas, guava is found in the wild as well as cultivated [182].

India is a major guava-producing country, accounting for approximately 56% of the total global production, followed by China, Thailand, Mexico, and Indonesia [2]. Similarly to other fruit crops, guava has multiple local names; it is called jambu batu in Malaysia, abas in Guam, and bayabas in the Philippines. The French refer to guava as goyave or goyavier, the Portuguese as goiaba, or goaibeira, and the Dutch as guyaba or goeajaaba [182]. Owing to their pleasant flavor, aroma, and nutritional content, guava fruits are well received by consumers.

Guava is susceptible to various fungal diseases, with wilt, fruit rot, dieback, styler end rot, stem canker, and fruit canker being the major diseases (Table 1). Among these, wilt is the most serious one, leading to substantial economic losses, particularly in India.

### 3.1. Wilt Disease

Guava wilt was first discovered in Taiwan in 1926 and later in the Allahabad district, India, in 1935 [191]. Most of the studies on guava wilt are from India, as the disease is considered a national problem owing to the economic importance of this fruit crop. The wilt disease has also been reported in several other guava-producing countries, including Pakistan [192], Bangladesh [193,194], South Africa [195] and Australia [196] (Table 1).

Partial wilting is a typical wilt of guava characterized by initial wilting of one side of the plant or some parts of the plants; later, other parts of the plant are also affected [58]. Guava wilts are categorized as quick and slow wilts. The quick-wilt-affected plants take approximately 2 weeks to 2 months to completely wilt after the appearance of initial symptoms. In a study by Misra and Pandey [53], guava plants with quick wilt required a minimum of 16 days for complete wilting. In the slow-wilt-affected plants, complete wilting takes approximately one year or more.

Guava wilt symptoms emerge after the rainy season, from October to November [197]. The fruits on infected plants remain undeveloped, hard, and stony. The infected guava plants appear yellow, with slight leaf curling, which leads to drooping and shedding of the leaves. Bark splitting can also occur in the wilted plants. Guava wilt infects both young and older trees bearing fruits, but older trees are more susceptible [198].

Comprehensive studies on the causal pathogens of guava wilt have been performed primarily in India. Various fungal pathogens, including *Fusarium oxysporum*, *Fusarium solani*, *Fusarium pallidoroseum*, *Fusarium decemcellulare*, *Fusarium equiseti*, *Gliocladium roseum*, *Gliocladium virens*, *Gliocladium penicilloides*, *Acremonium* sp., *Acremonium restrictum*, *Curvularia lunata*, *Curvularia pallescens*, *Chloridium virescens*, and *Pestalotiopsis dissieminata*, have been reported to be associated with guava wilt [53]. However, pathogenicity tests have indicated that only two species of *Fusarium*, *Fusarium oxysporum* and *Fusarium solani*, and two species of *Gliocladium*, *Gliocladium roseum*, and *Gliocladium penicilloides*, are the causal pathogens of guava wilt [53]. Prasad et al. [51] established *Fusarium oxysporum* as the causal pathogen of guava wilt and proposed the name *Fusarium oxysporum* f. ps *psidii* as a special form of *Fusarium oxysporum*. To date, the name *Fusarium oxysporum* f. sp. *psidii* is used when referring to the wilt disease of guava. *Fusarium solani*, in combination with *Microphomina phaseoli*, could also instigate the wilt of guava. Although more *Fusarium solani* isolates have been recovered from the roots of wilting guava and more have been found to be prevalent in the field, *Fusarium oxysporum* f. sp. *psidii* was found to be the most virulent. A study indicated that *Fusarium oxysporum* f. sp. *psidii* is the primary pathogen of guava wilt [52].

Two other *Fusarium* species, *Fusarium proliferatum* and *Fusarium chlamydosporum*, were found to be associated with this disease. These two species have been detected in severely affected guava orchards in India [52,199]. Pathogenicity tests were only performed on *Fusarium chlamydosporum* and confirmed it as a pathogen of guava wilt. Recently, Gangaraj et al. [59] reported *Fusarium oxysporum* f. sp. *psidii* and *Fusarium falciforme* as the causal pathogens of guava wilt in India.

### 3.2. Guava Decline

Decline is another serious guava disease that has been reported in India, Pakistan, and Brazil (Table 1). Guava decline is a complex disease that involves a synergistic interaction between *Fusarium* spp. and nematodes. During the synergistic interaction, infection of the root by nematodes predisposes the guava tree to *Fusarium* infection, thereby causing root rot [60,200]. The synergistic interaction between nematodes and *Fusarium* spp. was supported by a greenhouse and field assessment [201]. Two *Fusarium* spp., *Fusarium solani* and *Fusarium oxysporum*, and several species of nematodes are commonly associated with guava decline.

An inoculation study of nematodes and several root fungal pathogens of guava, including *Fusarium*, by Aftab [60] indicated that root-knot nematodes initiate root infection, and that the severity of decline symptoms could be due to the activity of fungal pathogens causing root rot. The role of nematodes is not only to allow fungi to enter the roots through wounded sites but also to alter the physiology of the root as well as the entire guava plant [202].

Guava decline symptoms are similar to wilt symptoms, which include chlorosis, wilting, leaf browning, and leaf drop. Noticeable symptom of guava decline is infected roots showing numerous galls, mainly due to infection by nematodes. Decline symptoms start appearing from the upper part of the infected tree and progress downwards [61,200]. Guava decline is the main disease of guava trees in Brazil, particularly in Rio de Janeiro. Initially, Gomes et al. [61] demonstrated that *Meloidogyne mayaguensis* and *Fusarium solani* cause the decline disease in guava in Brazil. Later, Gomes et al. [200,203] reported that the species complex of guava decline disease interacts with *M. enterolobii*, which parasitizes guava trees and allows infection by *F. solani*, thereby causing rotting of the roots.

Recently, Veloso et al. [64] reported that the decline disease of guava in Brazil is also caused by *Meloidogyne enterolobii* and *Fusarium solani*. According to Gomes et al. [61], the root knot nematode *Meloidogyne enterolobii* is a weak pathogen, and guava decline only develops with infection by *Fusarium solani*. A survey by Madhu et al. [62] in Haryana, India, revealed the presence of *Meloidogyne incognita* and *Fusarium oxysporum* and showed that their synergistic interaction contributes to guava decline incidence and severity. Decline was more severe in old orchards than in young orchards. Singh [63] reported that guava decline in Ratlam, India, is associated with the wilt fungus of guava, *Fusarium oxysporum* f. sp. *psidii*, and *Meloidogyne enterolobii*. Numerous galls were observed on the roots of infected guava trees. The same study indicated that *F. oxysporum* f. sp. *psidii* can cause both wilt and decline in guava. However, a detailed study is required to confirm the actual fungal pathogen of guava decline in the district of Ratlam, India, as the pathogen was identified based on mycoflora study.

Guava decline is also a major problem in Pakistan, as the disease develops rapidly and causes severe infection in many guava orchards. Guava decline in Pakistan is caused by various fungal pathogens, including *Fusarium solani*, *Fusarium oxysporum*, *Botryodiplodia theobromae* (synonym *Lasiodiplodia theobromae*), *Colletotrichum gloeosporioides*, *Helminthosporium* spp., and *Curvularia lunata*, as well as an oomycete, *Phythopthora parasitica* [65].

### 3.3. Anthracnose

Guava fruit rot is a serious disease that occurs in the field and during storage, transportation, and marketing. Various fungal pathogens can cause guava fruit rot in the field, as well as after harvest [58]. In this review, we focus on *Colletotrichum* which causes anthracnose (Table 1). The most obvious symptoms of anthracnose in guava are necrotic and sunken lesions on the surface of the fruits (Figure 2). Masses of spores are formed on lesions during the advanced stage of infection. Anthracnose not only infects the fruits but also the leaves and twigs of guava.

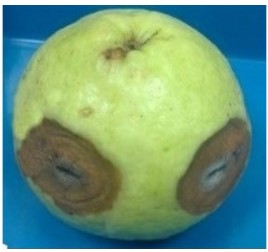

**Figure 2.** Anthracnose of guava.

Prior to taxonomic reevaluation of the genus *Colletotrichum*, *Colletotrichum gloeosporioides sensu lato* was considered the most common species associated with guava anthracnose in several guava-producing countries [204–207]. After identification based on the phylogeny of multiple markers, several species of *Colletotrichum gloeosporioides* and *Colletotrichum acutatum* complexes have been reported as the causal pathogens of guava anthracnose. Species of the *Colletotrichum gloeosporioides* complex have been reported as pathogens of guava anthracnose in Italy [66] and those of the *Colletotrichum siamense* complex in India and Mexico [67,68]. Within the *Colletotrichum acutatum* complex, so far, three species have been reported as the causal pathogens of guava anthracnose, *Colletotrichum abscissum* in Brazil [69], *Colletotrichum simmondsii* in Brazil [70], and *Colletotrichum guajavae* in India [71].

### 3.4. Crown Rot

Guava crown rot, caused by *Fusarium verticillioides*, has been reported in India [72]. The disease was detected in the Thai guava variety, which is an introduced variety in India. Guava crown rot usually occurs at the cut ends and spreads to the crown area, with discolored spots of various sizes [74]. Later, a larger lesion appears and turns into necrotic black circular patches that cover the whole fruit. Whitish mycelia appear in the lesions, and the fruits become discolored and malformed [72]. In a study by Valentino et al. [74], guava crown rot was reported to be caused by *Aspergillus* spp., *Aspergillus fumigatus*, *Aspergillus niger*, *Aspergillus tamarii*, *Aspergillus japonicus*, and *Aspergillus flavus*.

### 3.5. Fruit Rot

In addition to crown rot, another post-harvest disease of guava is fruit rot. Similarly to other post-harvest diseases, fruit rot of guava can occur in the field or during harvesting, storage, transit, and marketing. Similar to anthracnose, guava fruit rot lesions appear as brown spots, which later coalesce and become larger and expand to the whole fruit.

Several fungal species have been reported as the causal pathogens of guava fruit rot (Table 1). However, in some of the studies, the pathogens were identified based on morphological methods, and there is a possibility that the species identity of the pathogens is not accurate.

*Fusarium* spp. have been isolated from the fruit rot lesions of guava, among which *Fusarium oxysporum* has been reported in three localities in Egypt [77] and in several markets in Nigeria [76–78]. In a study by Amadi et al. [76], *Fusarium oxysporum* was found to be the most prevalent species isolated from guava fruit rot-infected plants. However, Latiffah et al. [208] reported that the *Fusarium oxysporum* and *Fusarium semitectum* isolated from guava rot-infected plants are not pathogenic.

Other guava fruit rot pathogens include *Aspergillus awamori* [79], *Phytophthora nicotianae* [80], *Neoscytalidium dimidiatum* [81], and *Lasiodioplodia theobromae* [82]. Furthermore, other post-harvest fungi recovered from guava rot lesions are *Pestalotia psidii*, *Rhizopus stolonifer*, *Aspergillus niger*, *Penicillium expansum*, *Rhizoctonia solani*, *Fusarium* sp., *Colletotrichum gloeosporioides*, *Fusarium oxysporum*, *Mucor* sp., *Rhizopus stolonifer*, *Aspergillus fumigatus*, and *Aspergillus parasiticus* [75,77].

### 3.6. Canker

The canker of guava affects the stems and fruits. The symptoms on the stem include longitudinal cracks and discoloration of the bark, which develop into large vertical cracks. The causal pathogen is *Diplodia natalensis* [83]. A common fruit canker is caused by *Pestalotia psidii*. The initial symptoms are small, brown, circular lesions. In severe infections, raised spots develop in large numbers, and the fruits may break, thereby exposing the seeds [83].

## 4. Fungal Diseases of Passion Fruit

Passion fruit (*Passiflora edulis*) is a vine belonging to the family Passifloraceae, which originated in southern Brazil through Paraguay to northern Argentina [182]. It is cultivated for commercial purposes in many tropical and subtropical regions. The local names of the fruit crops include buah susu (Malaysia), chum bap (Vietnam), linmangkon (Thailand), and lilikoi (Hawaii). In Spanish, passion fruit is called granadilla, parcha, parchita, parchita maracuyá, or ceibey, and in Portuguese, maracuja peroba [209].

The two major economically important varieties of passion fruit are the purple-type (*Passiflora edulis* Sims.) and yellow-type (*Passiflora edulis* f. *flavicarpa* O. Deg) [210]. Fruits of the purple type are often consumed fresh, while fruits of the yellow type are used in juice processing and preservation [182]. Passion fruits are categorized as minor tropical fruits as they are produced and traded at small amounts. Brazil is the major producer, followed by Peru, Colombia, Ecuador, Australia, New Zealand, Indonesia, and several African countries [2]. The major diseases of passion fruit caused by fungi are Fusarium wilt, root and collar rot, anthracnose, and scab (Table 1).

### 4.1. Vascular Wilt

Vascular wilt is an important disease of passion fruit, which can cause substantial yield losses. The disease is caused by *Fusarium oxysporum* f. sp. *passiflorae* and was first detected in Australia in 1950. Subsequently, vascular wilt was reported in several countries, including Brazil, Colombia, Korea, New Zealand, Panama, South Africa, Uganda, Venezuela, and Zimbabwe [87–89,211].

Vascular wilt affects both purple- and yellow-type passion fruit plants as well *Passiflora cincinnata*, which bears blue or violet flowers [86] and *Passiflora mollissima* [212]. According to Melo et al. [86], the incidence, severity, and mortality of wilt are higher in *Passiflora edulis*, the purple-type passion fruit. *Fusarium oxysporum* infecting passion fruit is host-specific, and the pathogenic strain produces distinctive proteins or effectors known as Secreted in Xylem (*SIX*), which may confer host-specific virulence [87].

Initial symptoms of vascular wilt include slight wilt at the branch tips and, sometimes, partial wilt or wilting of one side of the plant can occur. Thereafter, the entire plant wilts, followed by sudden death within 4–14 days [213]. Vascular tissues of the roots and lower stem appear dark brown to discolored. Cracks can also develop at the stem base [85]. As the fungus is a soil-borne pathogen, wilt usually starts in localized areas and spreads throughout the field via conidial dispersal. Conidia are often produced at high humidity [213].

*Fusarium solani* has also been isolated from the vascular tissues of *Passiflora edulis* showing wilting symptoms. When *Fusarium solani* and *Phytophthora nicotianae* v. *parasitica* were co-inoculated on *Passiflora edulis*, wilt developed rapidly, indicating that the concomitant infections of these pathogens can lead to the development of severe wilt [89]. According to Hirooka et al. [214], vascular wilt disease manifests as sudden wilt, and the authors recovered reddish perithecia from lesions at the collar part of the plant. Two pathogens, namely *Haematonectria ipomoeae* (=*Fusarium solani* f. sp. *melongenae*) and *Fusarium striatum*, were isolated from the infected roots, flowers, stems, and fruits of *Passiflora edulis* in Colombia.

## 4.2. Collar Rot

Collar rot caused by *Fusarium solani* is another severe disease of passion fruit. Bueno et al. [96] proposed *Fusarium solani* f. sp. *passiflorae* as the causative pathogen of collar rot; as in a phylogenetic tree based on the ITS region and EF-1α sequences, these isolates formed a distinct cluster from other formae speciales of *Fusarium solani*. Although there are fewer reports of collar rot than of wilt disease, the former can also cause significant yield losses and has been detected on passion fruit cultivated in Brazil, USA, China, Uganda [92–96,98].

Generally, collar rot affects passion fruit plants 1–2 years after planting, although infection may occur earlier in previously affected planting areas [215]. Initial symptoms include slight wilting, with leaf color turning pale green, followed by severe wilting and defoliation. Infected plants die when severe necrosis or necrotic girdling occurs in the infected areas. At high humidity, perithecia may emerge in infected collar tissues [96,216]. In addition to *Passiflora edulis* and *Passiflora edulis* f. *flavicarpa*, the disease affects several passion fruit species, including *Passiflora alata*, *Passiflora ligularis*, *Passiflora maliformis*, and *Passiflora quadrangularis* [95,96,216].

*Fusarium solani* can survive in the soil as chlamydospores for many years; thus, collar rot pathogen can spread through the transfer of infested soil or seedlings. Moreover, high temperature and humidity are conducive to the development of collar rot [96].

## 4.3. Canker

Symptoms of passion fruit canker are similar to those of vascular wilt, in which the leaves wilt and become chlorotic. However, plants infected with this disease develop concave cankers on the stem. Typically, however, defoliation does not occur, and the fruits remain on the plant. Infected tissues at the base of the stem may often be girdled at the soil line, which may be associated with root rot, adventitious root growth, and stem swelling at the canker site. Canker can be observed at the soil line or where the plant is strapped to the trellis. In infected stem tissues, perithecia are formed, and the infected plants often die within 5 years [99,100].

Two major pathogens causing passion fruit canker are *Fusarium solani* and *Fusarium oxysporum* f. sp. *passiflorae* [101,217]. According to Ploetz [217], *Fusarium solani* is not aggressive, and the presence of wounds plays a critical role in disease development. Wounding accelerates symptom appearance and increases disease severity. Through molecular identification, *Fusarium oxysporum* and *Nectria haematococca*-type isolates were detected in diseased tissues [101].

Passion fruit canker, reported as Nectria canker, has been detected in Taiwan and Uganda [92,102]. Another disease similar to Nectria canker, reported as the base rot of passion fruit, has been reported in subtropical Australia, but the causal pathogen was *Fusarium solani* [218].

Detailed reports on passion fruit canker are available from Florida [93]. To date, however, there has been no report of canker from other producing countries, due perhaps to the resemblance of its symptoms to those of vascular wilt, which may lead to description of the disease as wilt or dieback and collar rot, which also present similar symptoms. However, in a pathogenicity test, canker symptoms take a longer time to develop, indicating the "subtle nature" of the disease [93,101].

## 4.4. Anthracnose

Similar to other fruit crops, passion fruit is infected by anthracnose in the field or post-harvest during storage, transportation, and marketing. In addition to fruits, anthracnose affects leaves and twigs, causing defoliation and twig wilt [216]. Symptoms on leaves and twigs manifest as circular or irregular brown spots with dark edges [219]. On fruits, typical anthracnose symptoms are observed, starting as brownish-black spots, which co-

alesce to produce large sunken lesions with spore masses. Initially, most reports on passion fruit anthracnose indicated *Colletotrichum gloeosporioides* as the causal pathogen [217,220]. However, Tarnowski and Ploetz [108] reported four additional species, namely *Colletotrichum boninense*, *Colletotrichum truncatum*, *Colletotrichum gloeosporioides*, and *Glomerella* sp. as the pathogens of passion fruit anthracnose in Florida.

Based on molecular markers, several other species have been reported to be the causal pathogens of passion fruit anthracnose. For instance, *Colletotrichum boninense* has been identified as an anthracnose pathogen in Brazil [109], *Colletotrichum queenslandicum* in Northern Territory, Australia [110], *Colletotrichum brevisporum* in Fujian Province, China [111], *Colletotrichum capsici* (=*Colletotrichum truncatum*) in China and Taiwan [112,113], *Colletotrichum brasiliense* in China [114], and *Colletotrichum constrictum* in Yunnan, China [115].

### 4.5. Brown Spot and Septoria Spot

Brown spot and Septoria spot are the two other common diseases that infect passion fruit leaves and fruits. Two *Alternaria* species, namely *Alternaria passiflorae* and *Alternaria alternata*, are the causal pathogens of brown spot. Brown spot lesions caused by *Alternaria passiflorae* are larger than those caused by *Alternaria alternata*. Severe infection of leaves causes defoliation and fruit infection, which in turn reduces the quality and commercial value of the fruits [215,216].

The causal pathogens of Septoria spot or blotch are *Septoria fructigena*, *Septoria passifloricola*, and *Septoria passiflorae*. Leaves are the most susceptible to *Septoria* spp., although the disease also infects young twigs, flowers, and fruits. The latest report of *Septoria* spot was in Taiwan, which infected 2–3-month-old grafted passion fruit seedlings, and the causal pathogen was *Septoria passifloricola* [221].

## 5. Fungal and Oomycete Diseases of Lychee

Lychee or litchi (*Litchi chinensis* Sonn.) belongs to the family Sapindaceae, or the soapberry family. Presumably, lychee originated in Kwangtung and Fukien in southern China and has been cultivated for thousands of years in southern Guangdong [222]. From the 17th century onward, lychee cultivation spread to neighboring countries, particularly in Thailand, India, and Myanmar, and subsequently, in the 19th century, it expanded to the East Indies, Florida, and California [182].

In Asia, China is the leading producer of lychee, followed by India and Vietnam. Lychee production in China and India is mainly for the domestic market, particularly in China, as the fruit is very popular. Lychee produced in Vietnam is exported to China, the USA, Japan, and Australia. Following Asia, Africa is the most important region of lychee cultivation, with Madagascar being the leading producer of this fruit. Lychee cultivated in Madagascar is mainly exported to Europe, particularly France and the Netherlands [2].

Most of the published reports on the fungal diseases of lychee include anthracnose; pepper spot; and leaf, panicle, and fruit blight (Table 1). Other diseases include brown blight and tree decline [223]. Misra and Pandey [224] listed a number of fungal diseases of lychee in India, including leaf spot, root rot, and fruit rot, but the authors indicated that the diseases did not lead to a major economic impact on the yield.

The anthracnose of lychee infects not only fruits but also leaves, flowers, and flower stalk, leading to fruit rot, flower drop, and leaf spot. Infection is prevalent in warm and wet weather. The reported causal pathogen is *Colletotrichum gloeosporioides* and, sometimes, *Colletotrichum acutatum* [117]. Anthracnose of lychee fruits manifests as browning of the pericarp, but the fleshy part is typically unaffected [116]. Pericarp discoloration affects the appearance and quality of fruits. In Mexico, *Colletotrichum gloeosporioides* was isolated from immature fruits and asymptomatic flowers, indicating that the infection is a latent pathogen, and the pathogen may even be an endophyte [118]. In a recent study on lychee anthracnose, *Colletotrichum fioriniae* was found to be the causal pathogen of fruit

anthracnose [119] and *Colletotrichum karstii* was found to be the causal pathogen of leaf anthracnose [119].

Another disease affecting lychee is pepper spot was reported in Australia, Taiwan, and China. Symptoms of pepper spots appear as slightly raised, dark small spots on leaves, petioles, and fruits. On fruits, small spots may merge and cover much of the fruit surface. Similar to anthracnose, pepper spots also affect the appearance and quality of lychee fruits [225]. According to Anderson et al. [122], lychee pepper spots may also occur in other producing countries, such as the USA, Taiwan, and China, as symptoms similar to those of pepper spot have been reported.

*Colletotrichum gloeosporioides* has been reported as the causal pathogen of pepper spots in Australia [121,122] and the same pathogen also causes lychee anthracnose. In Taiwan and China, *Colletotrichum siamense* is the causal pathogen of lychee pepper spots [123,124].

The occurrence of leaf, panicle, and fruit blight of lychee caused by *Alternaria alternata* has been reported in Bihar, India [125]. Leaf blight occurs on old senescent leaves in mature lychee orchards and is regarded as economically non-important. Panicles and fruit blight appear during flowering and fruit development. Infected panicles become shriveled and dried, leading to necrosis. Pedicel infection causes necrosis, further leading to fruit blight and drying of the developing fruit. In addition to anthracnose, fruit rot caused by *Alternaria* sp. is a common post-harvest disease of lychee in India [125].

*Colletotrichum* has been recently reported to affect ripe lychee fruit in Hainan, China. The disease was detected in a field in which brown-to-black lesions were observed on the inner pericarp [126].

In addition to fungal diseases, downy blight caused by the oomycete *Phytophthora litchi* is a major disease of lychee. Downy blight infects young leaves, flowers, fruits, panicles, and shoots. Infected tissues became brown and are covered with masses of sporangiophores and sporangia. Oomycetes also cause fruit rot, resulting in considerable post-harvest losses [127,128].

## 6. Fungal and Oomycete Diseases of Longan

Longan (*Dimocarpus longan* Lour.) belongs to the family Sapindaceae, the same family as lychee, and is also called eyeball or dragon's eye. The fruit crop originated in southern China, specifically in Fukien, Kwangsi, Kwangtung, and Schezwan provinces [182]. Longan is commercially cultivated in China, Thailand, Cambodia, India, and Vietnam, with China and Thailand being the largest producers. Other producing countries include Bangladesh, Australia, South Africa, Reunion, Brazil, and Israel [226]. Longan production has increased over the past few years due to rising demand in China and Thailand. Moreover, Thai longans are preferred in China because of their superior quality, and import by China increased to approximately 140% in 2017 [2].

Over the years, there have been a substantial number of reports and publications on the fungal and oomycete diseases of longan. Diseases in the field affect the yield, and diseases on the fruits affect the commercial value of the commodity. Diseases in the field include downy blight, dieback, inflorescence wilt, and blight. Regarding fruits, pericarp browning, brown rot, and fruit rot have been reported (Table 1).

Downy blight of longan is caused by *Phytophthora litchi* (formerly known as *Peronophythora litchi*). The disease affects young leaves, panicles, flowers, and fruits. Obvious signs and symptoms of downy blight include infected tissues that become brown and covered with masses of white sporangia and sporangiophores [227]. *Phytophthora litchi*-infected leaf and stem rot of longan seedlings has been reported in Taiwan. The infected young leaves appear as droopy, blighted leaves, which eventually wither and fall. Water-soaked lesions are formed on infected leaves [129].

Another species, *Phytophthora palmivora*, also infected young shoots, panicles, and fruits. Necrosis appears in young shoots and irregular lesions appear on fruits, causing fruit rot and premature fruit drop [117,228]. *Phytophthora palmivora* was reported to cause

the brown rot of longan fruit in Thailand. Small brown spots appear on the skin, which develop into larger patches. Severe brown rot infection leads to fruit drop [131]. In addition to *Phytophthora litchi* and *Phytophthora palmivora*, which cause fruit rot, *Lasiodiplodia theobromae* and *Lasiodiplodia pseudotherobromae* have also been reported as the causal pathogens of longan fruit rot. *Lasiodiplodia theobromae* infected longan fruit in the field in Puerto Rico [132]. *Lasiodiplodia pseudotheobromae* was reported to cause longan fruit rot in Lamphun Province, Thailand, which developed post-harvest under high humidity and temperature [133].

Other fungi that infect longan fruit include *Phomopsis longanae* and *Lasiodiplodia theobromae*, causing pericarp browning, which affects the shelf life of fruits [134,135]. Pericarp browning is a major post-harvest disease of longan fruit in southern China [134].

Inflorescence and flower wilt and vascular necrosis of longan have been detected in Puerto Rico, and the causal pathogen was identified to be *Fusarium decemcellulare*. Inflorescence and flower wilt appeared in 50% plants and vascular necrosis on 70% plants [130]. Other diseases reported in Puerto Rico include dieback caused by four species of Botryosphaeriaceae, namely *Lasiodiplodia hormozganensis*, *Lasiodiplodia iraniensis*, *Lasiodiplodia pseudotheobromae*, and *Lasiodiplodia theobromae* [136]. *Lasiodiplodia theobromae* is also the pathogen of inflorescence blight of longan, which causes the rotting of flowers, rachis, and rachilla [132].

## 7. Fungal and Oomycete Diseases of Durian

Durian (*Durio zibethinus* L.) is a tropical fruit crop in the family Malvaceae, with the center of origin in Peninsular Malaysia, Indonesia, and Borneo [229]. From these countries of origin, durian was introduced to other Southeast Asian countries, including Thailand, Vietnam, the Philippines, and Myanmar. Durian is planted on a small scale in Hawaii, Costa Rica, Brazil, Ecuador, and Panama [230]. Thailand is the main exporter of durian, followed by Malaysia and Indonesia, and China is the leading importer of this fruit [2].

Durian is known as the king of tropical fruits because of its distinctive appearance, odor, and taste. Apart from the word durian from the Malay word "duri," meaning a thorn, other names or terms have been given to the fruits based on its smell, including stinkvrucht (stink fruit) in Dutch and civet fruit in India [182]. Durian is called tu-rien in Thailand, sau rieng in Vietnam, and liu-lian guo in Mandarin [230].

Economically important and the most severe diseases of durian are caused by the oomycete, *Phytophthora palmivora*, causing various types of diseases that infect durian in all growing countries. This oomycete is a soil-borne pathogen, and it can therefore infect the aerial parts of durian, leading to the infection of all plant parts and at all stages of plant growth. Moreover, hot and humid conditions as well as high rainfall are favorable for the pathogen growth and development [231].

The most severe diseases caused by *Phytophthora palmivora* are patch canker or stem canker, fruit rot, seedling dieback, foliar blight, and root rot [137–139]. The pathogen enters plant parts through natural openings or wounds [232]. *Phytophthora palmivora* is an effective pathogen, as the oomycete can reproduce rapidly under favorable conditions, and the zoospores are readily released from the sporangia in the presence of water [233]. Moreover, the pathogen produces several structures, including zoospores, sporangia, and chlamydospores for infection and spread [138].

The initial symptoms of patch canker caused by *Phytophthora palmivora* include distinct wet patches on the bark. When the patches on the stem or branch merge, the canker is formed, and reddish/brown substances ooze from the canker lesion. The lesion commonly spreads into the xylem, leading to leaf wilting and chlorosis and, ultimately, dieback [149]. Patch canker is regarded as the major disease of durian in Malaysia and was reported on many durian trees in Brunei, Darussalam [140]. Moreover, the incidence of patch canker was high in most durian orchards in Vietnam, and severe infection was noted on trees older than 20 years of age [141].

Symptoms of stem rot are rather similar to those of patch canker, in which the disease affects the trunk and bark and also causes yellowing and wilting of leaves. Stem rot is a severe disease of durians in Thailand. The causal pathogens are *Fusarium solani* and *Lasiodiplodia pseudotheobromae*, in addition to *Phytophthora* sp. [142].

In addition to *Phytophthora palmivora*, fungal pathogens cause durian leaf blight. In Vietnam and Peninsular Malaysia, *Rhizoctonia solani* is the causal pathogen of durian leaf blight [143,144]. Leaf blight occurs in nurseries and orchards, infecting individual leaves or the entire foliage. In orchards, when the symptoms are visible, the infection has spread to other parts of the tree [139,151], rendering control difficult. Leaf blight starts as small spots on leaves, which subsequently form larger lesions. The blighted lesions dry-up, turning dark brown, and the leaves appear necrotic and shriveled; the infected leaves easily fall, particularly when the disease is severe. Leaf blight reduces photosynthesis, which affects flower and fruit development.

Another fungal leaf disease is leaf spot caused by *Phomopsis durionis*, which infects durian seedlings and mature trees. Durian leaf spot is believed to be a latent infection in the field. Leaf spot begins as dark brown spots with a yellow halo, which are more obvious on mature leaves. The disease is more prevalent in poorly managed orchards, and the affected plants appear unhealthy due to reduced photosynthesis [145].

Symptoms of fruit rot caused by *Phytophthora palmivora* appear as small brown water-soaked patches on the outer skin, which later turn into dark brown or black lesions. Whitish mycelia and sporangia appear in the lesion. Infection can occur in both ripe and unripe fruits [234]. The rot lesion spreads to the pulp and seed, which affects the marketability of fruits as they are inedible [138]. Preharvest fruit rot of durian results in post-harvest rot, although the symptoms may not be obvious during or post-harvest. Infection can also occur when healthy fruits come in contact with infected fruits or orchard soil containing the pathogen inoculum. Infection of unwounded fruits can occur under prolonged (72 h) exposure to high humidity (minimum of 98%) [149]. Fruit rot can result in 10–25% loss [139].

Fungal pathogens can also cause durian fruit rot, causing symptoms similar to those caused by *Phytophthora palmivora*. *Sclerotium rolfsii* was found to be the pathogen of durian fruit rot that had come in contact with orchard soil [146]. Following harvest, several fungal pathogens were isolated from rot lesions of the infected fruit. *Colletotrichum gloeosporioides* and *Lasiodiplodia theobromae* are the most prevalent fruit rot pathogens [147]. Other pathogens including *Aspergillus* spp., *Penicillium* sp., and *Fusarium equiseti* are the secondary invaders or weak pathogens [148].

Durian tree decline has been reported in Queensland, Australia, and Indonesia. The disease was described to cause necrosis of the feeder root cortex tissues, rapid dieback, infrequent stem canker, and finally death. From infected durian trees in Queensland, Australia, two species of oomycetes, namely *Pythium vexans* and *Pythium palmivora*, were isolated, and the percentage of oomycetes isolated differed during dry and wet seasons. In both seasons, the percentage of *Pythium vexans* was higher than that of *Pythium palmivora*. A plant parasitic nematode, *Xiphenema* sp., was also isolated, suggesting a synergetic association between the oomycetes and nematode [149]. In Indonesia, diseases caused by *Phytophthora palmivora*, *Pythium cucurbitacearum*, and *Pythium vexans* led to durian tree decline [150].

In addition to *Phytophthora palmivora*, several species of *Phytophthora* and *Pythium* have been linked to durian diseases. *Phytophthora nicotianae* has been occasionally recovered from root rot and canker lesions [151]. Furthermore, *Phytophthora cinnamomi*, *Pythium cucurbitacearum*, *Pythium vexans*, and *Pythium deliense* have been reported as the causal pathogens of root and stem rot in durian trees [152,153].

*Phytophthora palmivora* remains a significant pathogen of durian and can infect a broad range of hosts, particularly wounded plants. Cross-infectivity studies have shown that *Phytophthora palmivora* isolates can infect other plant hosts. *Phytophthora palmivora* isolates from durian were found to be moderately pathogenic to papaya [235]. In addition,

isolates from cocoa could infect durian, rubber, oil palm, and coconut [236–239]. These studies indicate the possibility of field infection by *Phytophthora palmivora* in susceptible plants. The ability of *Phytophthora palmivora* to infect a wide range of hosts might be due to the presence of multiple pathogenic strains of this oomycete [239]. Additional more virulent or aggressive strains of *Phytophthora palmivora* may emerge. Therefore, disease surveillance of susceptible crops is essential to avoid severe outbreaks. Moreover, quarantine measures should be implemented on susceptible crops and planting materials brought into the country.

## 8. Fungal Diseases of Rambutan

Rambutan (*Nephelium lappaceum* L.) belongs to the family Sapindaceae, the same family as longan and lychee. The fruit is native to Malaysia and cultivated throughout Southeast Asia, with Indonesia being the largest producer, followed by Thailand [2,182]. Rambutan is also cultivated on smaller scales in the American tropics, South Africa, Hawaii, Puerto Rico, Sri Lanka, the Philippines, Australia, and Madagascar [182].

Rambutans are primarily used for domestic consumption. In the international market, the fruit remains a niche for Asian ethnic consumers in Europe and the US; and is also in demand from specialty or premium stores. Previously, rambutans were not well-known to Western consumers, and high retail prices restricted fruit marketing. However, retail prices of rambutans have dropped, and the demand for specialty fruits has risen; thus, rambutans have the potential to grow and gain more prominence in the Western market [2,239].

Several fungal diseases of rambutans, either in the field or post-harvest, have been reported. The common diseases include fruit rot, corky bark, stem canker, dieback, and powdery mildew (Table 1). These diseases have been reported in several rambutan-producing countries worldwide.

### 8.1. Fruit Rot

Fruit rot is one of the most severe diseases of rambutan, involving various fungal pathogens that infect fruits in the field and post-harvest. Infection in the field has the potential to affect fruits post-harvest, particularly if the fruits are not treated and stored under appropriate conditions [160], which ultimately affects their marketability.

Symptoms of fruit rot in mature and immature fruits are characterized by light to dark brown and sometimes black areas with water-soaked lesions on the surface of fruits. The lesions developed into the pericarp, causing blackening and drying of the infected parts (Figure 3A,B). The pericarp may crack, exposing the flesh [159,164].

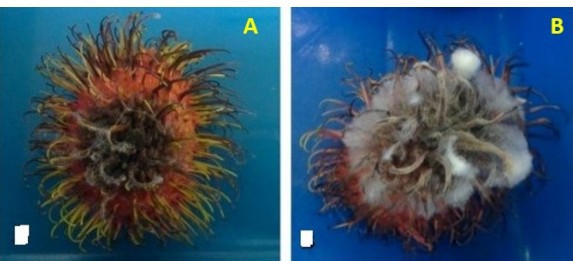

**Figure 3.** Rambutan fruit rot. (**A**) Fruit rot lesion. (**B**) Mycelia growth from the lesion.

Rambutan fruit rot has been reported in Hawaii, Puerto Rico, Malaysia, Thailand, and Sri Lanka. In previous reports of rambutan fruit rot in Thailand and Sri Lanka, various fungal pathogens or a complex of fungal pathogens were isolated from infected fruits, including *Botryodiplodia theobromae*, *Colletotrichum gloeosporioides*, *Gliocephalotrichum bulbilium*, *Pestalotiopsis* sp., *Phomopsis* sp., and *Glomerella* sp. [155,157]. In China, fruit rot of rambutan is referred to as a gray spot caused by *Pestalatiopsis* sp. [158].

In Hawaii, the causal pathogens of rambutan fruit rot include *Lasmenia* sp., *Gliocephalotrichum* spp., and *Pestalatiopsis virgatula* [159,160]. Several species of fungi have been recovered from fruit rot lesions, namely *Gliocephalotrichum bulbilium*, *Gliocephalotrichum simplex* [161]; *Calonectria hongkongensis* [162], *Colletotrichum fructicola*, and *Colletotrichum queenslandicum* [149]. In Malaysia, rambutan fruit rot is caused by *Gliocephalotrichum bacillisporum* [164].

### 8.2. Corky Bark, Stem Canker, and Dieback

Corky bark is a severe disease of rambutan and is often associated with stem canker and dieback [45,47,136]. *Dolabra nepheliae* is the causal pathogen of corky bark on rambutan, which was first detected in Malaysia [45]. Years later, the pathogen was reported on diseased rambutans in Hawaii, Puerto Rico, and Honduras. Corky bark initially emerges as irregular patches on the main stem and lateral branches, which later develop into brown lumps. The pathogen spreads to young stems or twigs, and the lumps appear as corky and rough structures bulging from the bark. As the stem and branches develop, the lumps or corky structures became larger, and the bark cracks, forming canker on the stem and branches. The canker formed on the stem appears roughened with irregular to spherical shapes. Ascomata of the pathogen develop in the bark fissure [166]. In severe cases, dieback of the branches manifests, which reduces tree growth [167].

The stem canker of rambutan was reported in Hawaii, Puerto Rico, and Honduras, and the causal pathogen was *Dolabra nepheliae*. The disease is relatively common in Hawaii and Puerto Rico and has likely been introduced from infected germplasm. Symptoms of stem canker normally appear on rambutan trees that are approximately 3 years old. The pathogen also infects branches. The bark fissures of infected stems appear swollen, with dark brown to black discoloration. As the tree matures, the size of the canker increases and the branches weaken and break due to the weight of the fruits, causing substantial fruit loss and tree damage [168,169].

In Puerto Rico, corky bark and dieback of rambutan caused by Botryosphaeriaceae species, including *Lasiodiplodia brasiliensis*, *Lasiodiplodia hormozganensis*, *Lasiodiplodia iraniensis*, *Lasiodiplodia pseudotheobromae*, *Lasiodiplodia theobromae*, *Neofusicoccum batangarum*, and *Neofusicoccum parvum*, were reported. The symptoms begin with necrosis of vascular tissues and branches, followed by dieback, in which the infected branches turned dark brown. The disease later spreads to the leaves. Corky bark symptoms appear, with the development of pycnidia on branches and the main stems, which become discolored [130].

### 8.3. Powdery Mildew

Powdery mildew of rambutan is caused by *Oidium nephelii*, an obligate parasite that is prevalent in Sri Lanka, the Philippines, Thailand, and Malaysia [][170,172]. The fungus appears as mycelia on young leaves in white patches, and subsequently completely covers leaves, flowers, and fruits. Infected fruits become discolored, turn black, and may crack, affecting fruit quality [172,240].

### 8.4. Other Fungal Diseases

*Pseudocercospora nephelii* was identified as the causal pathogen of leaf necrosis in rambutan seedlings, which can also lead to defoliation. Necrosis on leaves starts as small light brown spots that merge and become larger, turning 70–80% of the leaves necrotic. Severe infections can lead to defoliation and death of infected seedlings. The disease was reported in Brunei as well as in Sabah and Selangor in Malaysia [173]. Thus far, there has been no updated or latest report on the occurrence of this disease.

Inflorescence wilt and flower and vascular necrosis of rambutan were reported in Puerto Rico. The diseases affected 50% of rambutan inflorescences, and the causal pathogen was *Fusarium decemcellulare* [130], which also caused the same diseases on longan and mango in Puerto Rico.

### 9. Fungal Diseases of Mangosteen

Mangosteen (*Garcinia mangostana* L.) belongs to the family Guttiferae and is planted for fruits. The fruit crop is called the queen of fruits because of its sweet and sour taste, and it is mainly consumed fresh. Mangosteen is native to Southeast Asia, and the place of origin is believed to be the Sunda Islands and the Moluccas (Maluku Islands). In Kemaman, Malaysia, wild mangosteen trees are found in forests [182]. Southeast Asian countries, namely Thailand, Malaysia, Indonesia, and the Philippines, are the major mangosteen producers, with Thailand being the largest producer and exporter [2]. Mangosteens are also planted in South America, tropical Africa, and northern Australia [241].

Unlike other tropical fruit crops, pre- and post-harvest diseases of mangosteen are not a major problem. Several fungal diseases have been reported to infect mangosteen, including leaf disease, stem canker, fruit rot, and tree decline [179] (Table 1). The causal pathogens were similar to those reported for other fruit crops.

Leaf blight caused by *Pestalotiopsis flagisettula* has been reported in Thailand. Pestalotiopsis leaf blight has also been reported in Malaysia and North Queensland [152]. Brown leaf spots and blotches were detected in Hakalau, Hawaii, and the causal pathogen was identified to be *Pestalotiopsis* sp. [175].

Several fungal pathogens have been identified as the causal pathogens of mangosteen fruit rot, which occur mainly during post-harvest. Many of these fungal pathogens are secondary or weak pathogens that infect fruits through wounds in the field during storage and transportation [148]. *Diplodia* fruit rot affected mangosteen in Thailand, and the causal pathogen was reported to be *Diplodia theobromae* [174]. Several fungal pathogens are linked with mangosteen fruit rot in the field, with *Lasiodiplodia theobromae* being the causal pathogen of black aril rot, *Phomopsis* sp. of white aril rot, and *Pestalotiopsis* sp. of soft aril rot [179]. *Phomopsis* sp. and *Pestalotiopsis* sp. also cause mangosteen storage rot [177].

Imported mangosteen fruits in Guangzhou, China were infected by *Gliocephalotrichum bulbilium*, the pathogen causing discoloration of the pericarp and rotting of the edible flesh [176]. *Gliocephalotrichum bulbilium* and *Graphium* sp. are also associated with mangosteen storage rot [177]. Another fungal pathogen causing mangosteen fruit rot is *Mucor irregularis*, isolated from samples obtained from several markets and supermarkets in Wujing Town, Shanghai [178].

A decline in mangosteen trees was reported on the southern coast of Bahia, Brazil. Infection starts from the roots and spreads to the entire tree, with symptoms of wilting, yellowing, blighting, and defoliation, eventually causing tree death [180]. Two species of *Lasiodiplodia*, namely *Lasiodiplodia theobromae* and *Lasiodiplodia parva*, have been reported to be the causative pathogens of the disease. Other diseases of mangosteen include brown root rot caused by *Phellinus noxius*, stem canker and dieback caused by *Pestalotiopsis* sp. and thread blight caused by *Marasmiellus scandens* [174].

### 10. Management of Fungal and Oomycetes Diseases

Integrated disease management (IDM) is recommended to manage fungal and oomycete diseases of minor tropical fruit crops. Common methods in IDM are combinations of cultural, chemical, and biological methods which are applied to manage a wide range of diseases at the same time. Among the cultural methods commonly used are healthy seeds or propagating materials, field sanitation, and proper irrigation/drainage. These methods allow low or non-damaging levels of disease occurrence [242].

Some of the fruit crops are propagated through seed (rambutan, mangosteen, and durian), cutting (dragon fruits), budding, and grafting (passion fruit, longan and lychee).

The use of infected planting materials readily transmits and spreads the disease. Thus, the use of disease-free planting materials is important to avoid disease occurrence [243].

Proper irrigation and drainage are essential to manage wilt, root rot, crown rot, foot rot, stem rot, and die back, particularly caused by *Phytophthora*, *Pythium,* and *Fusarium*. The movement of water increases disease spreading to healthy plants which can be controlled by good and proper drainage. In the production area or nursery, accumulation of water should be avoided. To prevent infection, plants should be planted in raised beds which create drier conditions at the base. The pathogen or inoculum in irrigation and drainage water either in the nursery or in the field can infect the foliar part as well as the trunk and stem through water splashing, causing leaf blight, stem rot, gummosis, and canker [244].

Minor tropical fruit crops are perennial, of which several disease cycles occur and produce inoculum that can remain dormant. Pruning and removal of infected and dead plant parts including branches, leaves, twigs, and mummified fruits are the easiest methods to reduce the inoculum. Removal of canker should be followed by fungicide application such as Bordeaux mixture which allows faster wound healing and reduces dormant inoculum [243].

Sanitation in the nursery and in the field includes removing diseased or infected plant parts which can prevent spreading of pathogens to healthy plants. Diseased leaves, twigs, branches, stems, trunks, and fruits are sources of inoculum, and to eliminate the inoculum/pathogen, these plant parts are destroyed by burning. It is advisable to remove infected plant parts as soon as disease symptoms appear. Cleaning and disinfection of farming tools are part of sanitation methods of which this method can minimize disease spreading [243]. In addition to field sanitation, regular monitoring of healthy trees for signs and symptoms of diseases is essential to treat infected trees and to prevent widespread infections.

The application of fungicides has a large impact in fruit crops' disease management; thus, the use of fungicides should be within the scope of IDM strategy [245]. Important criteria to consider when using fungicides are the proper timing of application, the suitable fungicides to be used to manage the disease and the duration of protection by the fungicide. Triazoles are systemic and curative fungicides which are only effective during early infections. Most strobilurin fungicides provide protection of approximately 21–28 days after infection [246]. It is highly advisable to read the label carefully to ensure recommended fungicide for a particular disease is used and for efficient as well as safe use of the fungicide.

Biocontrol agents are alternatives to fungicides due to their perceived safety levels and minimal impact on the environment. Moreover, the current trend is switching towards reducing the use of agrochemicals to more eco-friendly methods of plant disease management. Several commercial products containing biocontrol agents are available to be used against fungal pathogens. Among the products are Primastop, containing *Clonostachys rosea* (*Gliocladium catenulatum*) to control wilt, seed rot, stem and root rot; Trichoderma Viride Trieco, containing *Trichoderma viridae* against soilborne fungal diseases; and RootShield®, containing *Trichoderma harzianum* to manage root rot diseases caused by *Pythium*, *Fusarium*, *Rhizoctonia*, and *Cylindrocladium* [247].

After harvest, the fruits can become infected by a number of fungal pathogens which affect the quality of the produce. Infection by fungi can occur in the field, during harvesting, handling, grading, storage, transportation and marketing [248]. The anthracnose pathogen becomes latent after infection and symptoms appear after harvest. Other fungal pathogens can directly penetrate the fruit's skin through wounds or mechanical injury, causing fruit rot.

Washing of fruits is carried out after harvest to remove dirt, latex and stains as well as to improve the appearance of the produce. The most common sanitation wash is chlo-

rine which is considered inexpensive. Other than washing, other forms of physical methods for the treatment of fruit crops are hot water treatment, heat treatment and gamma radiation [249].

Several fungicides including dichloran, imazalil, sodium ortho-phenil phenate, and thiabendazole are used as post-harvest treatments against wide range of fruit rot pathogens. Application of the fungicides are usually as drenches, high and low volume sprayers as well as dipping and flooder. However, due to issues related to fungicide residues and pathogen resistance, other alternative chemicals such as generally recognized as safe (GRAS) salts, namely sodium benzoate, sorbic acid, propionic acid, and acetic acid are used as postharvest treatment of fruit crops. These GRAS salts are applied as edible coating [250]. In addition to GRAS salts, chitosan, mineral oil, essential oil and cellulose are can also be used as edible coating [251].

There is a range of disease management methods and post-harvest technologies available that would enable smallholders and large plantation owners to improve the quality of the fruit crops. For smallholder farmers, they may face budget constraints to adopt new methods especially for postharvest practices and technologies.

## 11. Conclusions

Minor tropical fruits contribute to the economic growth and livelihood of smallholder farmers and promote local food security. However, several diseases affect production, which directly impacts the income of farmers. In most cases, common diseases of minor tropical fruits are wilt, stem diseases, anthracnose, and fruit rot. Diseases originating in the field such as anthracnose may express after harvest, particularly during storage. Fruit rot commonly occurs during storage due to improper handling and storage conditions. Thus, the knowledge of these diseases and their causal pathogens is paramount for planning and strengthening management in the field and during storage, transportation, and marketing post-harvest, as some of these fruit crops are grown for export markets.

However, there have been fewer studies on the diseases of minor tropical fruit crops compared to major tropical fruit crops, except for diseases on dragon fruits, as this fruit is cultivated commercially in many parts of the world, including several countries in Central and South America, Asia, and Australia. The great demand for dragon fruits in the international market has also contributed to many studies conducted on this fruit. However, the trend has now improved due to the rise in global demand for several minor fruit crops, such as passion fruit, guava, lychee, and longan. In addition to diseases, other research areas are also important, including post-harvest handling, storage conditions, and transportation, as minor tropical fruits have short shelf-lives and can easily perish.

**Funding:** This research received no external funding.

**Institutional Review Board Statement:** Not applicable.

**Informed Consent Statement:** Not applicable.

**Acknowledgments:** The author would like to thank the School of Biological Sciences, Universiti Sains Malaysia colleagues, postgraduate and undergraduate students for their contributions in part of the study on diseases of tropical fruit crops.

**Conflicts of Interest:** The author declares no conflict of interest.

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
