# Peer review of "Fungal and Oomycete Diseases of Minor Tropical Fruit Crops"

_horticulturae, doi:10.3390/horticulturae8040323_

Round 1

Reviewer 1 Report

This review paper comprehensively summarized the diseases of minor tropical fruit crops. It is very informative, and will provide great help for researchers in related fields. The manuscript is long, but well written. I have some suggestions as following:

  1. There are many short paragraphs in the text. I suggest part of them may be combined.
  2. It will be perfect if authors can provide some pictures on typical symptoms of some important diseases for each fruit crop.
  3. I suggest “5. Fungal and Oomycete Diseases of Lychee”, “6. Fungal and Oomycete Diseases of Longan”, “7. Fungal and Oomycete Diseases of Durian”, and “9. Fungal Diseases of Mangosteen” are also organized according to the symptoms.
  4. In the Conclusions, authors can summarize the main fungal and Oomycete species, and related disease symptoms.
  5. Line 219-220 “Fusarium, which is responsible for fruit rot, and newly reported fungal pathogens” are not related to Anthracnose, which should be removed.

Reviewer 2 Report

This article can review about important disease of some minor tropical fruits, and is very useful to understand the situation about those disease. I think it is very well summarized.

To make the review more accessible to the reader, it would be nice to have some photos of representative disease of each plant. Also, there are few some comments from me on the attached file, so please consider them for revise.

Reviewer 3 Report

Dear Author,

The review provides a wide area of knowledge regarding the most harmful diseases of minor fruits tropical fruits caused by Micromycetes fungi and fungi-like organisms. That what i missed reading this manuscript was information concerning plant protection tools, which are used to protect plants and limit the pathogens spreading, all the more reason, such recommendations for farmers the author wanted to provide (abstract and introduction). The next weak point is very poor language. I recommend checking the manuscript by native speaker.

The rest of comments as follow:

It's useless to provide in the text regarding description of diseases occurring on fruits, informations concerning countries where they have been reported, cuz such data are given in the Table 1.

In the whole text there are many repetition like in the lines 211 and 210...it must be corrected, since makes text a bit confusing.

Line 15

Instead ,,formulation" would be better ,,develope"

Line 27

Skip ,, producing"

Line 37

Instead ,, minor tropical fruits" use ,,products"

Line 47

Use ,, all kind of these fruits" instead ''all minor tropical fruit crops "

Line 50

Use ,, their production" instead ,,the production of these crops "

Line 70

Use ,, fungal pathogens" instead ,, fungal diseases''

Line 71.

The disease do not infecting the fruits, or generally plants.. This is reaction on infection caused by some group of pathogens, so instead ,, disease infecting" use ,,disease occurring on..."

Line 73

Instead ,, manifesting" use ,,showing"

Line 76

Skip ,,producing"

Line 79

Begining from this line, in the all text of manuscript (including Table 1) use abbreviations in case of genera names. Only first time use the whole name (for exemple; Colletotrichum gloeosporioides then C. gloeosporioides)

Line 102

Skip ,,On the surface of the sunken lesion'' and combine: Symptoms on infected Hylocereus polyrhizus (=Hylocereus monacanthus) include brown circular sunken lesions and etiological symptomsin the form of white mycelia and orange sporodochia.

Line 164

Gliocladium, G. roseum, and G. penicilloides, are the causal pathogens of guava wilt?? First time I found such information that Clonostachys spp. (the current name of the genus) can be pathogenic for plants...This genus can be pathogenic but for another fungi.

I would like to receive this paper where author found such information.

Line 190

Instead ,, wilt symptoms" use ,,wilt ones''

Line 191

Instead ,,prominent" use ,, substantials"

Line 193

Skip ,,guava"

Line 363

Insted ,,produced" use ,, caused"
